# Drought Stress Responses in Arabica Coffee Genotypes: Physiological and Metabolic Insights

**DOI:** 10.3390/plants13060828

**Published:** 2024-03-13

**Authors:** Habtamu Chekol, Bikila Warkineh, Tesfaye Shimber, Agnieszka Mierek-Adamska, Grażyna B. Dąbrowska, Asfaw Degu

**Affiliations:** 1Department of Plant Biology and Biodiversity Management, College of Natural and Computational Sciences, Addis Ababa University, Addis Ababa 3434, Ethiopia; habtamu.chekol@aau.edu.et (H.C.); bikila.warkineh@aau.edu.et (B.W.); 2Ethiopian Institute of Agricultural Research, Addis Ababa 2003, Ethiopia; gessesetesfaye@yahoo.com; 3Department of Genetics, Faculty of Biological and Veterinary Sciences, Nicolaus Copernicus University in Toruń, Lwowska 1, 87-100 Toruń, Poland; mierek_adamska@umk.pl (A.M.-A.); browsk@umk.pl (G.B.D.)

**Keywords:** drought stress, *Coffea arabica*, growth, gas exchanges, metabolites, network analysis

## Abstract

Understanding the impact of drought stress on Arabica coffee physiology and metabolism is essential in the pursuit of developing drought-resistant varieties. In this study, we explored the physiological and metabolite changes in coffee genotypes exhibiting varying degrees of tolerance to drought—namely, the relatively tolerant *Ca*74110 and *Ca*74112, and the sensitive *Ca*754 and *Ca*J-19 genotypes—under well-watered conditions and during terminal drought stress periods at two time points (0 and 60 days following the onset of stress). The metabolite profiling uncovered significant associations between the growth and the physiological characteristics of coffee genotypes with distinct drought tolerance behaviors. Initially, no marked differences were observed among the genotypes or treatments. However, at the 60-day post-drought onset time point, notably higher shoot growth, biomass, CO_2_ assimilation, pigments, and various physiological parameters were evident, particularly in the relatively tolerant genotypes. The metabolite profiling revealed elevations in glucose, maltose, amino acids, and organic acids, and decreases in other metabolites. These alterations were more pronounced in the drought-tolerant genotypes, indicating a correlation between enhanced compatible solutes and energy-associated metabolites crucial for drought tolerance mechanisms. This research introduces GC-MS-based metabolome profiling to the study of Ethiopian coffee, shedding light on its intricate responses to drought stress and paving the way for the potential development of drought-resistant coffee seedlings in intensified agro-ecological zones.

## 1. Introduction

Coffee stands as a vital global agricultural commodity, trailing only behind oil in importance. Its production in tropical and subtropical regions sustains millions of livelihoods [1,2]. *Coffea arabica* L. accounts for over 70% of the world’s coffee production and is famed for its excellence [3]. Brazil leads in Arabica coffee production at 44%, with Ethiopia contributing 5% [4]. Ethiopia, a top Arabica coffee producer in Africa, exports about 471,000 tons yearly, yielding 0.71 tons per hectare [5,6,7]. The looming specter of global climate change, however, threatens *C. arabica* cultivation with water scarcity and drought. This poses significant challenges to coffee cultivation, disrupting suitable regions, yield, and quality, and inviting pests and diseases, causing economic losses [8,9].

Plants instinctively adjust their catabolic and anabolic systems during drought, altering metabolic pathways to protect against damage [10,11]. This adaptation’s mechanism hinges on the species, genotype, and stress intensity [12,13]. Metabolic adjustment involves accumulating compatible solutes, influencing pathways like sugar synthesis, photosynthesis, and more [14]. Certain metabolites increase during drought stress, like proline, serine, valine, and betaine, fostering tolerance [15,16], while others decrease, like myo-inositol and glutamate [17]. Metabolite accumulation helps in cell turgor maintenance, osmotic potential reduction, and oxidative damage protection [18]. Stress-resilient plants often maintain higher stress-related metabolite levels even under normal conditions [13,19]. Despite the importance of metabolomic components in drought tolerance, insights into *C*. *arabica*’s metabolomics under drought stress remain limited [20]. Understanding drought stress adaptation necessitates studying the metabolomic responses of drought-sensitive and tolerant genotypes [21].

Metabolomic analysis, utilizing techniques like nuclear magnetic resonance and liquid/gas chromatography–mass spectrometry, can precisely reveal the level of small-molecule metabolites [22,23,24]. Nevertheless, understanding coffee’s intricate genetic and molecular aspects remains a significant hurdle [2]. The recent focus on coffee has illuminated the metabolite responses to drought [2], temperature [19], elevated CO_2_ [21], and water logging [25]. However, previous research in Ethiopia has predominantly approached drought stress in coffee from an agronomic and yield perspective, with limited comprehensive metabolomic profiling. Given metabolites’ role in osmoregulation during drought stress, bolstering coffee drought research with metabolomic investigations is crucial. Thus, this study aimed to characterize and profile the metabolite response to drought in tolerant and sensitive *C*. *arabica* genotypes, enhancing our grasp of drought adaptation and possibly guiding the development of robust coffee varieties. By understanding these mechanisms, we may pave the way for more climate-resilient coffee genotypes.

## 2. Results

### 2.1. Shoot Growth and Biomass Were Affected by Drought Stress Treatments

Distinct variations in growth were evident among the four coffee genotypes following a 60-day period of drought-induced stress. When compared to the well-watered coffee genotypes, those exposed to drought stress exhibited a notably lower stem height, shoot fresh, and dry biomass at the end of the stress period, with statistically significant differences (*p* < 0.05). Within the conditions of drought stress, a markedly higher stem height (Figure 1), shoot fresh biomass (Figure 2A), and shoot dry biomass (Figure 2B) were observed in the relatively tolerant genotypes *Ca*74112 (18.16 ± 2.42 cm, 9.3 g, and 2.71 g) and *Ca*74110 (15.35 ± 1.19 cm, 7.11 g, and 2.01 g), whereas lower values were recorded in the sensitive genotypes of *Ca*754 (11.4 ± 1.3 cm, 4.43 g, and 1.17 g) and *Ca*J-19 (13.25 ± 1.19 cm, 5.73 g, and 1.57 g). In comparison to the well-watered genotypes, at the conclusion of the experiment under drought stress conditions, the relatively tolerant genotypes *Ca*74112 (38.33%) and *Ca*74110 (43.59%) displayed the smallest reductions in shoot growth, while the sensitive genotypes *Ca*754 (50.84%) and *Ca*J-19 (47.17%) experienced larger reductions in stem heights (Appendix A). Similarly, at the end of the experiment, the smallest reduction in both shoot fresh and dry weights was observed in the relatively tolerant genotypes *Ca*74112 (31.01%, 30.87%) and *Ca*74110 (41.34%, 41.23%), compared to the sensitive genotypes *Ca*754 (62.14%, 62.13%) and *Ca*J-19 (60.21%, 60.25%) (Appendix A).

### 2.2. Differences in Relative Water Content among Coffee Genotypes under Drought Stress

In well-watered conditions, there were no significant (*p* > 0.05) differences among the genotypes in terms of the relative water content, with values ranging between 81.16 and 82.76% (Figure 3A). However, at the end of the experiment, in the plants grown under the drought stress conditions, the mean relative water content was significantly (*p* < 0.05) lower than those in well-watered conditions and the value of RWC was different among the coffee genotypes, with higher RWC values identified in the relatively tolerant genotypes of *Ca*74112 (48.11 ± 0.9%, 41.89% reduction compared to well-watered (*ww*) conditions) and *Ca*74110 (43.40 ± 0.29%, 46.74% reduction compared to well-watered (*ww*) conditions), and lower RWC values were recorded in the sensitive genotypes of *Ca*754 (30.24 ± 0.21%, 62.74% reduction compared to *ww* conditions) and *Ca*J-19 (32.57 ± 0.13%, 60.32% reduction compared to *ww* conditions) (Appendix A).

### 2.3. Effects of Drought Stress on Stomatal Conductance among the Coffee Genotypes

At the initial stage, no significant differences in stomatal conductance (Gs, mmol m^−2^s^−1^) were observed between the plants grown under drought stress and well-watered conditions. However, by the end of the experiment, noticeable variations in Gs were evident across the genotypes. Among the drought-stressed plants, the highest Gs values were recorded in the relatively tolerant genotypes *Ca*74112 (60.27 ± 1.39 mmol m^−2^s^−1^) and *Ca*74110 (46.57 ± 0.9 mmol m^−2^s^−1^), while the lowest Gs values were recorded in the sensitive genotypes *Ca*J-19 (36.86 ± 0.72 mmol m^−2^s^−1^) and *Ca*754 (30.3 ± 0.87 mmol m^−2^s^−1^) (Figure 3B). Because of the imposed drought stress, there were reductions in Gs by 38.32%, 53.31%, 63.85%, and 69.34% in the *Ca*74112, *Ca*74110, *Ca*J-19, and *Ca*754genotypes, respectively (Appendix A).

### 2.4. Drought Stress-Associated Variation in Carbon Assimilation among Coffee Genotypes

The impact of drought stress on the net carbon assimilation rate (A_net_, µmol CO_2_ m^−2^s^−1^), stomatal conductance (Gs, mmol m^−2^s^−1^), and transpiration rate (E, mmol m^−2^s^−1^) were examined. There were no significant differences in the CO_2_ assimilation rate among the genotypes when grown under the control conditions. However, when subjected to drought stress conditions, all genotypes displayed distinct reductions in the CO_2_ assimilation rate, exhibiting a gradual decline throughout the experiment (Figure 3C). The relatively tolerant genotypes of *Ca*74112 (2.91 ± 0.12 µmol m^−2^s^−1^) and *Ca*74110 (2.31 ± 0.07 µmol m^−2^s^−1^) recorded higher A_net_ values, while lower values were observed in the sensitive genotypes *Ca*J-19 (1.57 ± 0.14 µmol m^−2^s^−1^) and *Ca*754 (1.02 ± 0.1 µmol m^−2^s^−1^). In assessing the impact of drought stress on the reduction inCO_2_ assimilation rate, the sensitive genotype *Ca*754 experienced the largest reduction (85.96%), significantly greater than the smallest reduction observed in the tolerant genotype *Ca*74112 (59.85%) (Appendix A).

### 2.5. Variations in Photosynthetic Pigments under Drought Stress among Coffee Genotypes

Across all the tested coffee genotypes, drought stress led to a notable decline in *Chl*-a (Figure 4A) and *Chl*-b (Figure 4B) content, while in well-watered plants, the chlorophyll levels remained relatively stable throughout the experiment. By the end of the drought stress period, genotype *Ca*74112 recorded the highest and lowest *Chl*-a and *Chl*-b values (1.09 mg g^−1^fw and 0.21 mg g^−1^fw, respectively), while *Ca*754 had the lowest values (0.63 mg g^−1^fw for *Chl*-a and 0.14 mg g^−1^fw for *Chl*-b) (*p* < 0.05). Comparing the reduction rates of *Chl*-a and *Chl*-b due to drought stress, the sensitive genotype *Ca*754 displayed the largest reduction (56.96% for *Chl*-a and 57.14% for *Chl*-b), whereas the tolerant genotype *Ca*74112 exhibited smallest reduction (28.56% for *Chl*-a and 38.72% for *Chl*-b) (Appendix A).

### 2.6. Alterations in Metabolites under Drought Stress Conditions

To investigate the molecular changes associated with drought tolerance, this study examined the metabolic responses at two distinct time points: 0 and 60 days into the drought implementation period. A profiling analysis identified 50 metabolites spanning sugars, amino acids, and intermediates from pathways such as the tricarboxylic acid cycle (TCA), glycolysis, γ-aminobutyric acid (GABA) shunt, and shikimic pathways. The relative concentrations of each metabolite are represented as fold changes (*ws*/*ww*) (see Figure 5, Appendix A, Appendix A). Initially, no significant differences in metabolite alterations were observed between relatively tolerant and sensitive coffee genotypes in both treatment groups. However, 60 days after drought stress begins (DADB), a noteworthy shift occurred. The relatively tolerant genotypes, *Ca*74112 (with 46 altered metabolites—30 up and 16 down) and *Ca*74110 (with 46 altered metabolites—29 up and 17 down), displayed significantly larger changes compared to the sensitive genotypes, *Ca*J-19 (42 altered metabolites—23 up and 19 down) and *Ca*754 (40 altered metabolites—22 up and 18 down). At the outset of the study (0TL), no significant metabolite alterations were evident between the well-watered/control (*ww*) and drought stress (*ws*) treatments. However, by 60 DADB, substantial alterations in metabolite accumulation were observed compared to the control conditions (*p* < 0.01). At 60 DADB, the relatively tolerant genotypes exhibited markedly larger alterations and increased metabolite accumulation compared to the sensitive genotypes. Specifically, in the relatively tolerant genotypes (*Ca*74110 and *Ca*74112), the most substantial accumulation was observed forglucose (68.67- and 93-fold), tryptophan (67- and 80-fold), maltose (42.4- and 42.27-fold), galactose (25.8- and 37.7-fold), L-cysteine (2- and 48.44-fold), lactose (27.12- and 32.87-fold), lysine (14.18- and 22.71-fold), methionine (20.4- and 18.15-fold), leucine (9.66- and 14.45-fold), pyruvic acid (13.06- and 12.83-fold), ribose (8.82- and 9.59-fold), and other amino acids and organic acids. Conversely, in *Ca*754, *Ca*J-19, *Ca*74110 and *Ca*74112, significant reductions (*p* < 0.05) were observed in glutamate (0.13-, 0.16-, 0.15-, and 0.15-fold, respectively), aconitic acid (0.65-, 0.56-, 0.37-, and 0.28-fold), glucose-6-phosphate (0.53-, 0.68-, 0.12-, and 0.33-fold), γ-aminobutyric acid (0.49-, 0.39-, 0.36-, and 0.48-fold), fructose-6-phosphate (0.74-, 0.81-, 0.29-, and 0.41-fold), asparagine (0.38-, 0.29-, 0.51-, and 0.57-fold), glutamine (0.23-, 0.22-, 0.19-, and 0.59-fold), aspartate (0.66-, 0.47-, 0.56-, and 0.63-fold), alanine (0.82-, 0.98-, 0.74-, and 0.64-fold), myo-inositol (0.69-, 0.68-, 0.67-, and 0.68-fold), and citric acid (0.55-, 0.27-, 0.63-, and 0.86-fold) among the four genotypes (see Appendix A, Appendix A).

### 2.7. Correlation between Specific Metabolites and Growth and Physiology

An interesting aspect of this study was to combine the knowledge from the growth and physiological responses of the coffee genotypes with the metabolic data. For this purpose, a correlation analysis of the growth and physiological parameters and metabolic responses at 60 DADB was performed. Among the highly increased sugar metabolites, strong positive correlations between glucose, maltose, and galactose and shoot fresh biomass (*r* = 0.89, *r* = 0.837, *r* = 0.927, respectively), RWC (*r* = 0.939, *r* = 0.942, *r* = 0.993, respectively), A_net_ (*r* = 0.946, *r* = 0.905, *r* = 0.964, respectively), and Gs (*r* = 0.884, *r* = 0.842, *r* = 0.936, respectively) were observed. Moreover, strong positive correlations between tryptophan, L-cysteine, and proline and shoot fresh biomass (*r* = 0.867, *r* = 0.886, *r* = 0.908, respectively), RWC (*r* = 0.969, *r* = 0.771, *r* = 0.987, respectively), A_net_ (*r* = 0.917, *r* = 0.805, *r* = 0.948, respectively), and Gs (*r* = 0.88, *r* = 0.891, *r* = 0.918, respectively) were observed. From the TCA cycle and glycolysis pathway, strong positive correlations between malic and pyruvic acid andRWC (*r* = 0.922, *r* = 0.968, respectively), A_net_ (*r* = 0.841, *r* = 0.923, respectively), and Gs (*r* = 0.808, *r* = 0.878, respectively) were observed. Strong negative correlations between myo-inositol and shoot fresh biomass (*r* = −0.613), RWC (*r* = −0.695), A_net_ (*r* = −0.713), Gs (*r* = −0.595), and E (*r* = −0.61) were observed. Glutamate was weakly correlated with shoot fresh biomass (*r* = 0.539), RWC (*r* = 0.426), A_net_ (*r* = 0.562), and Gs (*r* = 0.496). α-ketoglutaric acid and glucose-6-phosphate were negatively correlated with shoot fresh biomass (*r* = −0.966,*r* = −0.573, respectively), RWC (*r* = −0.933, *r* = −0.758, respectively), A_net_ (*r* = −0.933, *r* = −0.682, respectively), and Gs (*r* = −0.976, *r* = −0.584, respectively). The data for all the significant correlations among the growth and physiology parameters and metabolites are available in Appendix A.

### 2.8. Network Analysis of Metabolic Alterations

The 60 days of drought stress caused an increase in the number of edges, network density, average node degree, and other key values of the metabolite network analysis, and the increase was greater in the relatively tolerant genotypes compared to the sensitive coffee genotypes. Thus, at *r* > 0.8, the mean values of network analysis results in the relatively drought-tolerant genotypes showed that they had 87 nodes, 1250 edges, a 14.4 to 1 edge-to-node ratio, network density of 0.359, average node degree of 29.762, characteristics path length of 4.268, clustering coefficient of 0.924, network heterogeneity of 0.385, network diameter of 10, network radius of 5, and network centralization of 0.126, while the sensitive genotypes had 80 nodes, 1411 edges, a 17.6 to 1 edge-to-node ratio, network density of 0.907, average node degree of 36.293, characteristics path length of 1.094, clustering coefficient of 0.991, network heterogeneity of 0.211, network diameter of 3, network radius of 2, and network centralization of 0.071 (Figure 6 and Figure 7, Appendix A).

### 2.9. PCA Analysis

The PCA results demonstrate an obvious metabolite accumulation and genotype category during the drought stress exposition. The first principal component (PC1) and second principal component (PC2) represented 86.77% and 8.39%, respectively. PC1 was dominated by the metabolites, glucose, maltose, tryptophan, galactose, lactose, lysine, methionine, leucine, valine, ribose, isoleucine, pyruvic acid, and serine; maltose, fumaric acid, malic acid, ferulic acid, α-ketoglutaric acid, chlorogenic acid, valine, shikimic acid, and succinic acid were major contributors for separation along PC2 (Figure 8, Appendix A).

In order to understand the drought-responsive metabolites, the metabolites’ up- and down-regulation score values were considered for the analysis in each genotype, and the score value of PC1 (86.77%) was taken as the weight. As a result, the most responsive metabolite PC1 score values were for glucose (11.96 PCA score value), maltose (9.79 PCA score value), tryptophan (8.08 PCA score value), galactose (5.47 PCA score value), lactose (4.50 PCA score value), lysine (4.42 PCA score value), methionine (3.23 PCA score value), leucine (2.99 PCA score value), valine (2.94 PCA score value), ribose (2.71 PCA score value), isoleucine (2.70 PCA score value), pyruvic acid (2.70 PCA score value), and serine (2.60 PCA score value) (Appendix A).

Similarly, following the same protocol, to examine the drought tolerance capacity of the genotypes, the genotype vs. the water treatments score values were taken into consideration. The result showed that the PC score value of the genotypes under drought stress conditions was significantly (*p* < 0.05) higher than those under the well-watered conditions. However, among the coffee genotypes under drought stressed conditions, the highest PC1 score values were from the relatively tolerant genotypes of *Ca*74112 (0.972 PCA score value) and *Ca*74110 (0.977 PCA score value) which were followed by the sensitive genotypes of *Ca*J-19 (0.803 PCA score value) and *Ca*754 (0.679 PCA score value) (Appendix A, Appendix A).

## 3. Discussion

### 3.1. Genotypic Variability and Physiological Responses

The physiological impact of drought stress on plants is often manifested in restricted growth and developmental limitations due to the scarcity of water [26]. Coffee plants, both at the seedling or mature stages, exhibit high sensitivity to soil moisture levels, which profoundly affect their subsequent growth and development [7,27]. Consistent with the findings of Silva et al. [9], our study noted a significant decline in the growth performance of coffee plants under drought stress, reflected in a reduced shoot length and lower fresh and dry biomasses. Notably, after a 60-day drought stress period, a relatively higher shoot length and increased fresh and dry biomasses were observed in the relatively tolerant genotypes *Ca*74110 and *Ca*74112 compared to the sensitive genotypes *Ca*754 and *Ca*J-19. Under drought stress, any decrease in turgor pressure and water potential can impede cell division, expansion, and elongation, leading to a reduced leaf area, smaller leaf size, and ultimately lower photosynthetic rates by limiting CO_2_ assimilation [7,9,26]. Studies by DaMatta et al. [28] and Wei et al. [29] have that drought stress on diminishes shoot and root growth in coffee and *Lycium barbarum* plants, respectively. However, previous research by Chekol et al. [7] and Dias et al. [30] suggest that in tolerant genotypes, enhanced growth responses are linked to water conservation mechanisms that enable coffee plants to sustain cell division and elongation processes. Studies by Caine et al. [25] and Xiong et al. [31] on rice and oak plants, respectively, also support the notion that limited water availability triggers metabolic responses favoring cellular division. This, in turn, favors the development of dermal tissues, ground tissues, and vascular tissues, which are essential components contributing to the plant’s adaptation to drought stress. Shoot growth and development serve as key indicators of a plant’s response to drought stress and are often considered key parameters in assessing a plant’s drought tolerance [32]. Similar to the observations of Mirian et al. [33], our results showed that the relatively tolerant genotypes *Ca*74110 and *Ca*74112 maintained higher growth metrics—such as shoot length and fresh and dry biomasses—more effectively than the sensitive genotypes of *Ca*754 and *Ca*J-19 under drought stress conditions.

### 3.2. Relative Water Contents, Gas Exchange, and Pigment Variations among Coffee Genotypes under Drought Stress

In the current study, drought stress significantly impacted various physiological parameters of the coffee genotypes, notably reducing the leaf relative water content, net assimilation rate, stomatal exchange, and chlorophyll pigments compared to well-watered conditions. Similar reductions in these parameters under drought stress were reported in other studies on coffee [7,9], cowpea [34], and other tolerant crops [35]. The tolerant coffee genotypes (*Ca*74110 and *Ca*74112) displayed a higher relative water content even under drought stress, aligning with findings in potato genotypes reported by Soltys-Kalina et al. [36].

Drought stress often leads to decreased photosynthesis assimilation rates and stomatal conductance, and can affect gaseous exchange parameters [26,31]. Similarly, tolerant coffee genotypes exhibited better physiological performances in these parameters compared to sensitive genotypes under drought stress [7,30]. The reduction in photosynthesis rate under drought is usually associated with stomatal closure and decreased internal CO_2_ concentrations, impacting CO_2_ fixation and pigment synthesis [9,37,38]. This decline in photosynthesis rate, along with the reduced stomatal conductance, often leads to diminished pigment synthesis [39]. The tolerant genotypes (*Ca*74110 and *Ca*74112) maintained higher pigment contents even under drought stress compared to the sensitive genotypes (*Ca*754 and *Ca*J-19). Drought stress often affects the structural organization and functions of photosynthetic pigments by destroying thylakoid membranes and reducing the activity of essential enzymes like RUBISCO [40]. This stress-associated decline in chlorophyll content has been observed in various crops, indicating damage to light-harvesting complex proteins, impacting photon absorption and electron transport [41]. Thus, relatively drought-tolerant genotypes (*Ca*74110 and *Ca*74112) could possess better protective mechanisms against chlorophyll degradation enzymes than sensitive genotypes (*Ca*754 and *Ca*J-19).

### 3.3. Drought Stress Causes Variability in Metabolite Alterations among Coffee Genotypes

Drought stress triggers significant changes in the biosynthesis and transport of metabolites, orchestrating adjustments in plants’ physiological and biochemical processes [17]. Plants respond in diverse ways to the shifting soil moisture regimes, either showing tolerant or sensitive behaviors [9]. Tolerant plants usually sustain metabolic processes and defense responses, whereas sensitive ones operate in the opposite manner [42]. In our study, we focused on sugar, amino acid, and organic acid synthesis in different pathways like the glycolysis, GABA shunt, TCA cycle, and shikimic pathways. After a 60-day drought stress period, the relatively tolerant genotypes accumulated more metabolites (28 in *Ca*74110 and 29 in *Ca*74112) compared to the sensitive genotypes (25 in *Ca*754 and 21 in *Ca*J-19). Conversely, the sensitive genotypes down-regulated more metabolites (18 in *Ca*754 and 22 in *Ca*J-19) than the relatively tolerant ones (17 in *Ca*74110 and 16 in *Ca*74112). This reflects a significantly higher metabolite accumulation in the relatively tolerant *Ca*74112 and *Ca*74110 genotypes compared to sensitive *Ca*754 and *Ca*J-19 genotypes (*p* < 0.01). According to Fabregas and Fernie [20], organic biomolecules like sugars, amino acids, and others play pivotal roles in osmotic adjustment during drought stress by regulating the vacuolar osmotic potential. Kapoor et al. [43] also noted that these metabolite responses to drought vary not only between species but also among genotypes and different parts of the plant. The PCA analysis further revealed distinct separation and clustering between the relatively tolerant and sensitive coffee genotypes, indicating diverse mechanisms of metabolite regulation in response to drought stress. Previous findings by Rodrigues et al. [2] on coffee plants and Hochberg et al. [13] on grapevines suggest that prolonged drought stress triggers adjustments in various metabolites, enhancing the plants’ resilience to drought. Similarly, Xiong et al. [31] studied Quercus species and found that during drought stress, metabolite alterations are more pronounced in tolerant genotypes, providing enhanced resistance to manage plant growth and development.

Sugars are metabolite classes that are highly sensitive to drought stress and have been extensively studied [16]. The profiling of 12 sugars showed increased levels of these sugars across all genotypes under drought stress conditions, which were notably higher in the relatively tolerant *Ca*74110 and *Ca*74112, particularly for glucose, maltose, galactose, lactose, and ribose, compared to the sensitive *Ca*754 and *Ca*J-19. However, in the current study, the myo-inositol concentration decreased universally, indicating its drought sensitivity. These sugars play key roles in osmotic adjustments, membrane stability, and maintaining the leaf water content during drought stress [20]. The results of previous studies by Urano et al. [44], Krasensky and Jonak [45], Fabregas et al. [46], Ogbaga et al. [47], and Pires et al. [48] align with our findings, having demonstrated increased fructose, glucose, raffinose, and other sugar levels during drought stress in various plants. Additionally, similar to our observations, Urano et al. [44] also noted a reduction in myo-inositol content under drought conditions.

This study measures the levels of 19 amino acids, with certain amino acids showing an accumulation across all genotypes, which was particularly pronounced in the relatively tolerant *Ca*74110 and *Ca*74112; these included tryptophan, L-cysteine, lysine, methionine, valine, leucine, isoleucine, serine, and proline. However, alanine, aspartate, glutamate, glutamine, and asparagine levels decreased in all genotypes. During drought stress, amino acids usually act as osmolytes and scavengers of reactive oxygen species, thereby influencing cellular functions [49]. Proline accumulation correlates with drought tolerance [50,51], as demonstrated by various studies including Konieczna et al. [26], Zhang et al. [52], and Joshi et al. [53]. Decreases in certain amino acids during drought stress might be due to redirecting metabolic activities towards proline biosynthesis [54]. Enhanced protein degradation or inhibition of biosynthesis presumably contributes to increases inamino acid levels during prolonged drought [48]. Increases in amino acids linked to pyruvate metabolism might be due to their involvement in gluconeogenesis to alleviate transamination products [55]. The correlation-based network analysis also demonstrated heightened coordinated metabolic activities in the relatively tolerant coffee genotypes, showcasing their resilience to drought stress. Hochberg et al. [13]’s findings in grapevines support this, indicating that prolonged drought stress can boost the metabolic network density. Likewise, Sanchez et al. [56] observed increased network connectivity in lotus genotypes facing salt stress, aligning with these findings.

Among the identified eight tricarboxylic acid (TCA) cycle intermediates, mostly marked reduction levels across all genotypes were apparent, which were particularly higher in the relatively tolerant *Ca*74110 and *Ca*74112 (malic acid, oxalic acid) compared to the sensitive *Ca*754 and *Ca*J-19. The TCA cycle metabolite responses to drought stress are less pronounced than the sugar and amino acid responses. Similar to this study, Araujo et al. [57] also noted limited alterations to the TCA cycle during drought stress. Fabregas et al. [46] demonstrated analogous changes in Arabidopsis, linking increased malic acid and oxalic acid levels to suppressed malate dehydrogenase levels, aiding nutrient uptake and intracellular ionic regulation under drought conditions. The increase in malic acid and oxalic acid levels is associated with decreased sink tissue utilization due to malate dehydrogenase suppression [58]. Yang et al. [59] also reported declines in citrate, succinate, α-ketoglutarate, and fumarate levels in drought-stressed maize kernels, aligning with this study’s findings.

This study examined 11 metabolites from the glycolysis, GABA, and shikimic biosynthetic pathways, observing increased levels in all genotypes, which were more pronounced in the relatively tolerant *Ca*74110 and *Ca*74112 (pyruvic acid, shikimic acid) compared to the sensitive *Ca*754 and *Ca*J-19. However, fructose-6-phosphate, glucose-6-phosphate, γ-aminobutyric acid, succinate semialdehyde, and putrescine exhibited reduced levels across all genotypes. Consistent with this study, Rabara et al. [17] also reported decreased fructose-6-phosphate and glucose-6-phosphate levels in tobacco and soybean leaves under drought stress. Guo et al. [15] observed a declining trend in GABA shunt metabolites in wheat under drought. According to Kinnersley’s review [60], γ-aminobutyric acid (GABA) levels increased in response to drought stress in various plant species.

## 4. Materials and Methods

### 4.1. Plant Material

The study used four *C*. *arabica* L. (*Ca*) genotypes sourced from the Jimma Agricultural Research Center (JARC). The selection of these genotypes was based on their drought tolerance, comprising both relatively tolerant (*Ca*74112 and *Ca*74110) and sensitive (*Ca*754 and *Ca*J-19) genotypes, as previously reported by Chekol et al. [7] and Tesfaye [61]. Our experimental focus was to investigate the interplay between growth, physiological performance, and metabolite responses within adult coffee genotypes. Adhering to the guidelines from WCR [62], we transplanted germinated coffee plants once they exhibited the first leaf pair, and were disease-free, had 3–5 cm tall stems (hypocotyls), and 2–3 cm secondary roots. These germinants were transplanted into 5 L plastic pots, with the side of the pot covered with aluminum foil to prevent excessive heat buildup. The pots, with drainage holes, were filled with 4 L of mixed topsoil, compost, and sand (2:1:1 ratio, pH 5.4–6.8). To address specific nutritional requirements and align with distinct coffee growth stages, we added 2.0 g of NPK/DAP fertilizers 5–7 cm below the seedlings. Subsequently, uniform-looking seedlings of each genotype were placed within a greenhouse environment and received consistent watering prior to the initiation of the drought stress treatments (Appendix A).

### 4.2. Growth Condition and Experiment Design

The research was conducted within controlled greenhouse conditions where the relative humidity ranged between 50 and 70%, with an average temperature of 24.5 °C and a photon flux density of 850 ± 13 μmol m^−2^s^−1^. Coffee genotypes (240 days aged), with 7–8 leaf pairs and free from disease or nutrient deficiencies, were used for the study, and subjected to two conditions: well-watered (*ww*) and drought-stressed (*ws*). Under *ww* (soil moisture of 60–80% field water capacity), seedlings were irrigated every 3–4 days. In contrast, for the *ws* conditions, seedlings were initially fully irrigated to the same field water capacity and subsequently subjected to drought conditions by withholding water until the experiment’s end (around 300 days of coffee age). The study used a completely randomized block design (CRBD) with four genotypes and two water regimes, each replicated ten times, totaling 80 coffee plants.

To evaluate the coffee growth and physiological performance in response to drought stress, at 10-day intervals until the end of the experiment (around 60 days), shoot height, leaf relative water content, stomatal conductance, and net carbon assimilation rate were measured. Moreover, evaluations of pigments of the coffee genotypes were conducted at the beginning of the experiment (0 days) and end of the experiment (60 days after drought initiation).

For the metabolite analysis, we sampled the third matured leaf from upper new flesh growth at two distinct time points during the study: at the start of the experiment (0 days) and end of the experiment (after 60 days of the drought implementation period). These fresh biomass leaf samples were snap frozen in liquid nitrogen and stored at −80 °C for further metabolite analysis (Appendix A).

### 4.3. Leaf Relative Water Content

At 10-days intervals until the end of the experiment (around 60 days), the relative water content (*RWC*) from representative leaves of the coffee genotypes was calculated based on the following formula of Barrs and Weatherley [63]:(1)$$RWC=\frac{\left(FW-DW\right)}{\left(TW-DW\right)}\times 100$$
where *FW* represents the leaf fresh weight, *DW* represents the leaf dry weight, and *TW* represents the leaf turgid weight.

The leaves’ fresh weight was measured, and thenthe samples were soaked in distilled water for 2 h at room temperature (20–22 °C) and the turgid weight was determined. Furthermore, the samples were dried to a constant weight at 70 °C, and the dry weight was determined. Sample weights were measured using balance to an accuracy of 0.0001 g (Sartorius, Bangalore, India).

### 4.4. Gas Exchange Measurements

At 10-days intervals until the end of the experiment (around 60 days), instantaneous gas exchange measurements were measured periodically. The rate of stomatal conductance (Gs, mol H_2_O m^−2^s^−1^) and net carbon assimilation (A_net_, µmol CO_2_ m^−2^s^−1^) were collected using a LI-6400 open gas exchange system (LI-COR, Lincoln, NE, USA) adjusted at 400 μmol CO_2_ mol^−1^ air reference CO_2_ concentration, 1000 μmol m^−2^s^−1^ photosynthetic photon flux density, and 500 μmol s^−1^ flow rates. The measurements were conducted on a young and fully expanded leaf, between 9:00 and 11:00 a.m.

### 4.5. Content of Photosynthetic Pigments

For the pigment analysis (chlorophylls) [64], at 0 and 60 days after the start of the drought treatment, leaf discs were collected from healthy and fully expanded leaves which were used for gas exchange measurements, and the concentration of pigments were analyzed using a UV–VIS spectrophotometer (Model 3092, Maharashtra, India). The concentration of chlorophyll a (*chl-*a) and chlorophyll b (*chl*-b), were measured based on the following formulas:*Chl-a* (mg/g tissue) = 12.25 A_663.2_ − 2.79 A_646.8_(2)
*Chl-b* (mg/g tissue) = 21.50 A_646.8_ − 5.10 A_663.2_(3)
where *Chl*-a represents the content of chlorophyll a (mg g^−1^ tissue), and *Chl*-b represents the content chlorophyll b (mg g^−1^ tissue).

### 4.6. Vegetative Growth Measurements

At 10-days intervals until the end of the experiment (around 60 days), the stem height (SH, cm, using meter scale) growth performance in response to drought stress was measured. Following sample harvesting (plants at around 300 days of age), shoot fresh biomass (g) and dry biomass (g, oven-dried biomass at 70 °C for 24 h) were measured using a sensitive balance (0.0001 g accuracy, Sartorius, Bangalore, India).

### 4.7. Metabolite Analysis

To analyze metabolites, the dried leaf biomass samples (that were frozen using liquid nitrogen and stored at −80 °C) of the well-watered and drought stressed conditions from 0TL and 60TL were ground to a constant weight as a fine powder under liquid nitrogen using a mortar and pestle. The powder was oven-dried to a constant weight at 70 °C for a period of 24 h. Approximately 100 mg powder was weighed, and then extracted in a 1 mL methanol/chloroform/water extraction solution (2.5:1:1 *v*/*v*) [65]. The mixture was thoroughly vortexed (MX-S, Scilogex, Rocky Hill, USA) and kept in an orbital shaker (OS-20Pro, Joan Lab Equipment Co., Ltd., Huzhou, China) for a duration of 15 min. Following this initial preparation, the samples underwent centrifugation (MSLZL19, Neuvar, Palo Alto, CA, USA) for 10 min at 12,000 revolutions per minute (rpm) and placed at 4 °C. The resulting supernatant was then carefully transferred to 2 mL screw-top tubes, mixed with 300 μLof chloroform and 300 μLof mass spec (MS)-grade water, and then centrifuged at 20,000× *g* for 2 min. Subsequently, 100 μLof the polar phase (water–methanol phase) was dried in a vacuum concentrator (Vacufuge Plus, Eppendorf, Hamburg, Germany) at 30 °C for a period of 3 h, and stored at −80 °C.

The dried polar extracts were derivatized with 40 µL of 20 mg mL^−1^ methoxyamine hydrochloride, followed by 70 µL of *N*-methyl-*N*-trimethylsilyltrifluoroacetamide (TMS derivatization) and 20 µL mL^−1^ of a mixture of fatty acid methyl esters (FAMES). For the metabolite analysis, this study utilized gas chromatography–mass spectrometry (GC-MS), employing an Agilent 7890 system coupled with a DB-5MS capillary column coated with a 5% diphenyl and 95% dimethylpolysiloxane mixture. During injection, an aliquot of the analyte (1 μL) was injected in the splitless mode. Helium served as the carrier gas, with a specific temperature program ranging from 90 °C to 285 °C. Peaks were manually annotated, and ion intensity was determined, and the metabolites were identified through systematic comparison with an established reference library derived from the Golm Metabolome Database [66], based on retention time and indices, and mass spectra, enabling us to gain insights into the intricate metabolic profiles of the coffee genotypes under study. In order to understand the alterations in the metabolites, the resulting ion intensities were transformed and normalized, based on relative concentration, for the removal of measurement bias, and to prepare the data for further statistical analyses.

### 4.8. Statistical Analysis

Statistical analyses of the collected data were performed using the Analysis of Variance test in SigmaPlot version 13 (Systat Software Inc., San Jose, CA, USA). To identify significant differences among the experimental groups, post hoc multiple comparisons were performed using Tukey’s honest significant difference test (*p* < 0.05). The dataset was transformed in Past version 4.0.3 [67]. Pearson correlation analyses between all the metabolite pairs and among the metabolite, growth, and physiological traits was performed after checking the assumptions of normality using the Shapiro–Wilk test. To reconstruct a metabolite network that would capture the coordinated changes in the metabolic profiles, threshold values were determined. Network visualization of metabolites was performed using Cytoscape version 3.10.1 [68], and the number of edges, number of nodes, edge to node ratio, network density, average node degree, characteristics path length, clustering coefficient, network heterogeneity, network diameter, network radius, and network centralization were investigated. To construct correlation-based networks of significant correlations, *r* > 0.8 threshold values were applied. Principal component analyses were performed on the transformed value of *ws*/*ww* using RStudio (version 4.2.1).

## 5. Conclusions

This study explored the impact of drought stress on Arabica coffee, focusing on the physiological and metabolic changes in genotypes with varying tolerance to drought. Notably higher shoot growth, biomass, CO_2_ assimilation, and pigments were evident in tolerant genotypes (*Ca*74110 and *Ca*74112) compared to the sensitive group under drought stress. The metabolite profiling revealed elevated levels of glucose, maltose, amino acids, and organic acids, suggesting an increase incompatible solutes is crucial for drought tolerance; these changes were more pronounced in the drought-tolerant genotypes. In this context, delving further into gene expression presents a promising avenue for the development of drought-tolerant coffee genotypes aimed at achieving sustainable yields and productivity.

## Figures and Tables

**Figure 1 plants-13-00828-f001:**
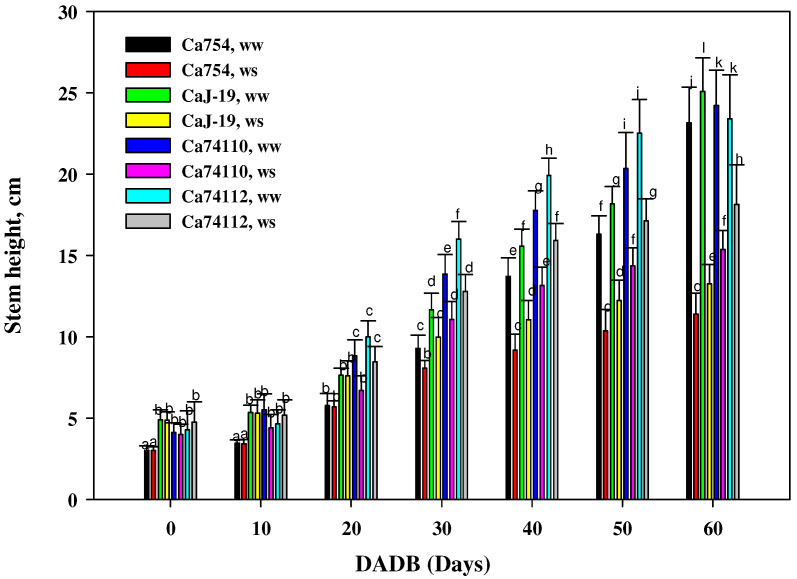
Stem length responses of *Ca*754, *Ca*J-19, *Ca*74110, and *Ca*74112 coffee genotypes under drought stress (*ws*) and well-watered (*ww*) conditions at different DADB (days after drought stress begins). Bars (means ± SD, *n* =10 replicates per genotype) with the same letter indicate no significant difference.

**Figure 2 plants-13-00828-f002:**
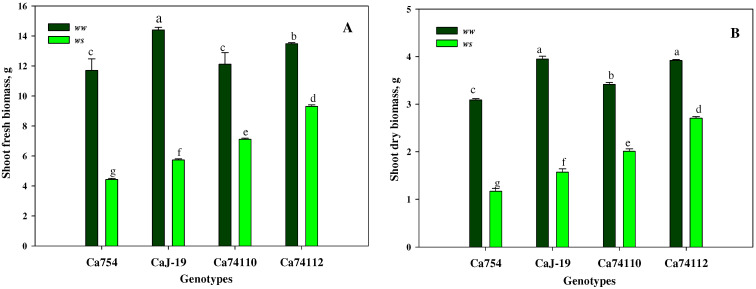
Shoot (**A**) fresh biomass and (**B**) dry biomass responses of *Ca*754, *Ca*J-19, *Ca*74110, and *Ca*74112 coffee genotypes under drought stress (*ws*) and well-watered (*ww*) conditions, 60 days after drought stress initiation. Bars (means ± SD, *n* =10 replicates per genotype) with the same letter indicate no significant difference.

**Figure 3 plants-13-00828-f003:**
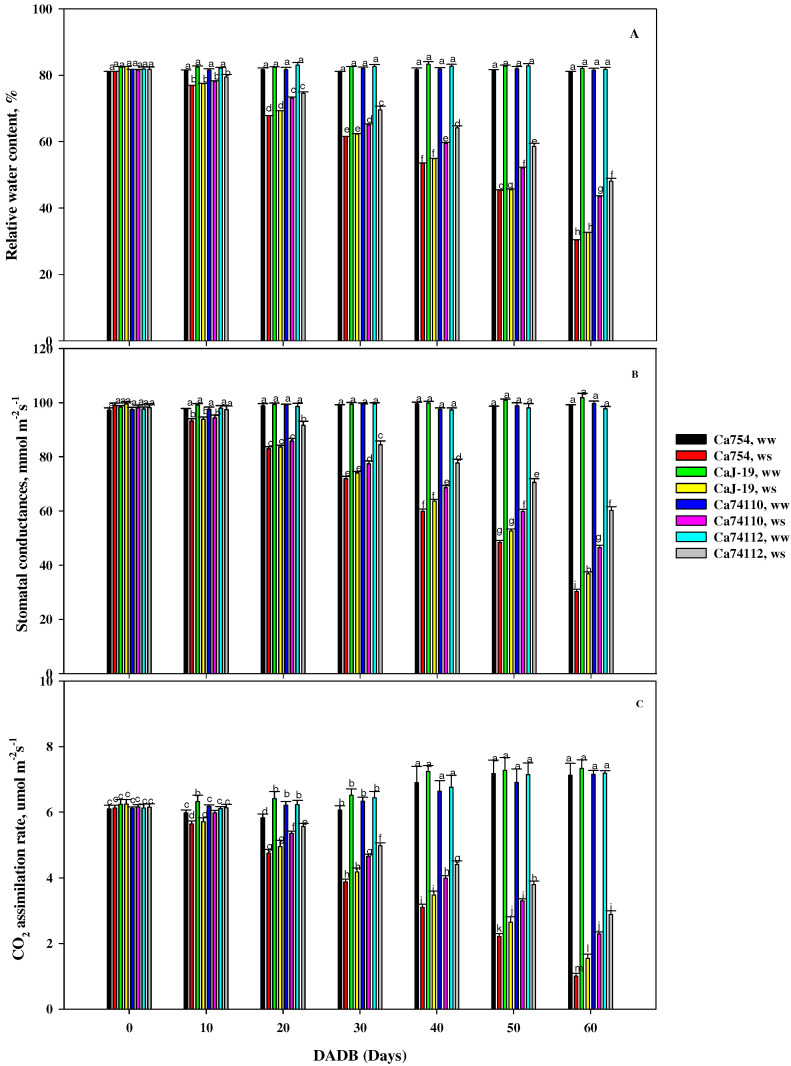
(**A**) Relative water content, (**B**) stomatal conductance, and (**C**) net CO_2_ assimilation rate of *Ca*754, *Ca*J-19, *Ca*74110, and *Ca*74112 coffee genotypes under drought stress (*ws*) and well-watered (*ww*) conditions at different DADB (days after drought stress begins). Bars (means ± SD, *n* = 10 replicates per genotype) with the same letter indicate no significant difference.

**Figure 4 plants-13-00828-f004:**
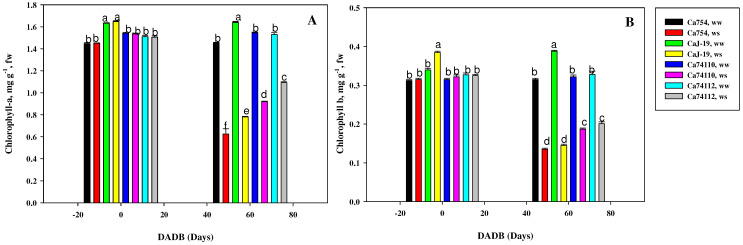
(**A**) Chlorophyll-a and (**B**) chlorophyll-b content of *Ca*754, *Ca*J-19, *Ca*74110, and *Ca*74112 coffee genotypes under drought stress (*ws*) and well-watered (*ww*) conditions at 0 and 60 DADB (days after drought stress begins). Bars (means ± SD, *n* =10 replicates per genotype) with the same letter indicate no significant difference.

**Figure 5 plants-13-00828-f005:**
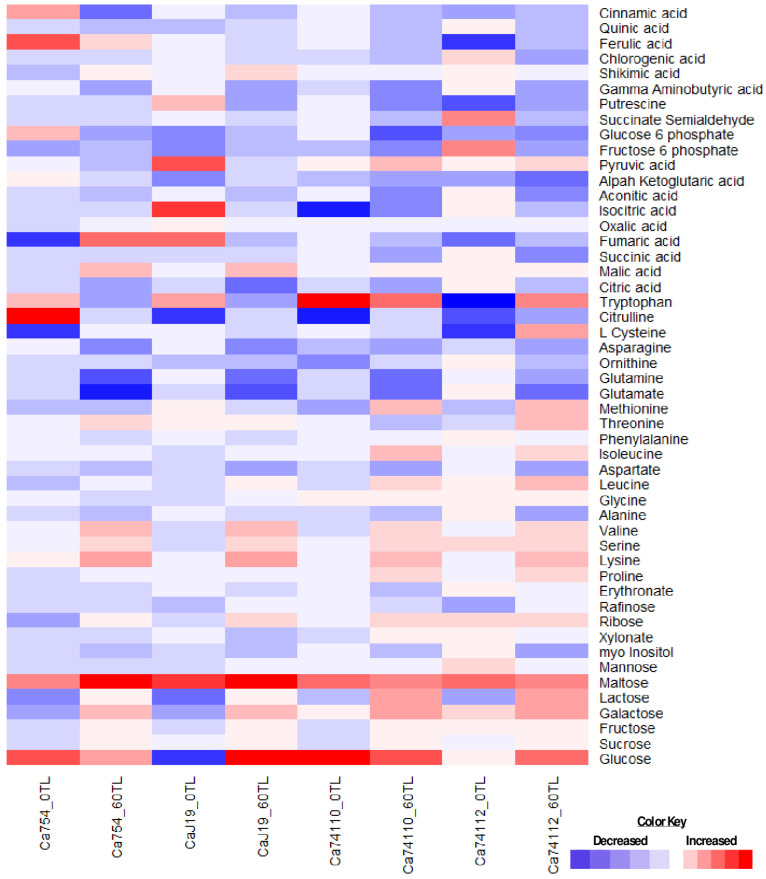
Metabolic responses to drought stress in leaves of *Ca*754, *Ca*J-19, *Ca*74110, and *Ca*74112 coffee genotypes. Each heatmap box represents logarithmic (Log_2_)-transformed fold change values (*ws*/*ww*, drought stressed/well-watered) of selected leaf metabolites at 0 and 60 days after drought stress initiation. Red, blue, and white represent an increase, decrease, and intermediate values, in terms of metabolite alteration. TL refers the experiment time line.

**Figure 6 plants-13-00828-f006:**
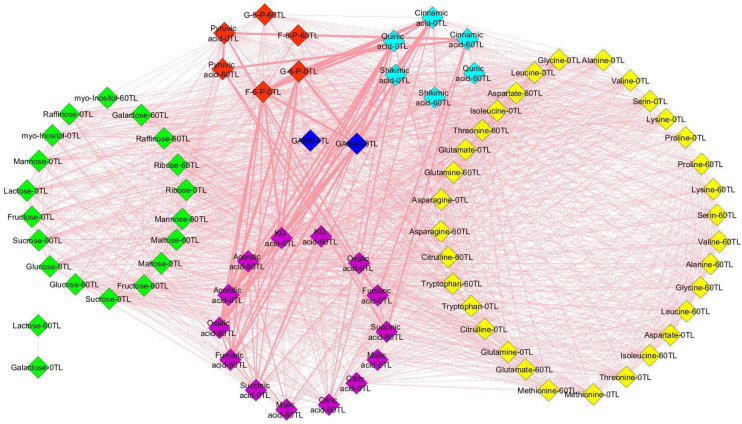
Changes in *Ca*754 and *Ca*J-19 metabolite interactions as a result of drought stress conditions at 0TL and 60TL. Nodes correspond to metabolites and edges between nodes represent Pearson correlations with *r* ≥ 0.8. Yellow, green, red, pink, blue and aqua colors represent amino acids, sugars, glycolysis pathway, tricarboxylic acid pathway, γ-aminobutyric acid pathway and shikimic acid pathway metabolite groups, respectively. TL refers to the time line.

**Figure 7 plants-13-00828-f007:**
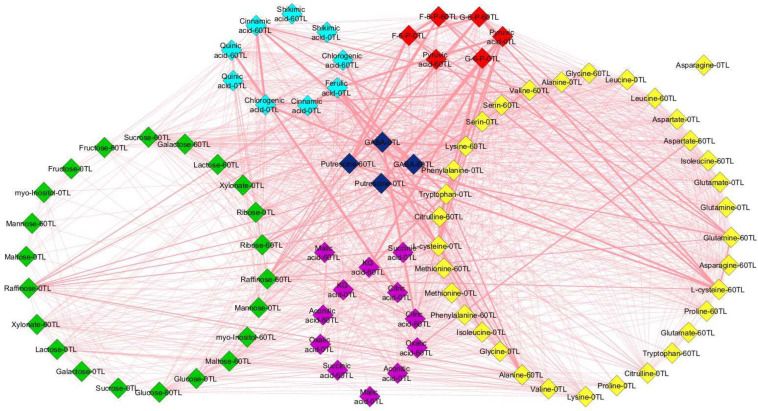
Changes in *Ca*74110 and *Ca*74112 metabolite interactions due todrought stress conditions at 0TL and 60TL. Nodes correspond to metabolites and edges between nodes represent Pearson correlations with *r* ≥ 0.8. Yellow, green, red, pink, blue and aqua colors represent amino acids, sugars, glycolysis pathway, tricarboxylic acid pathway, γ-aminobutyric acid pathway and shikimic acid pathway metabolite groups, respectively. TL refers to the time line.

**Figure 8 plants-13-00828-f008:**
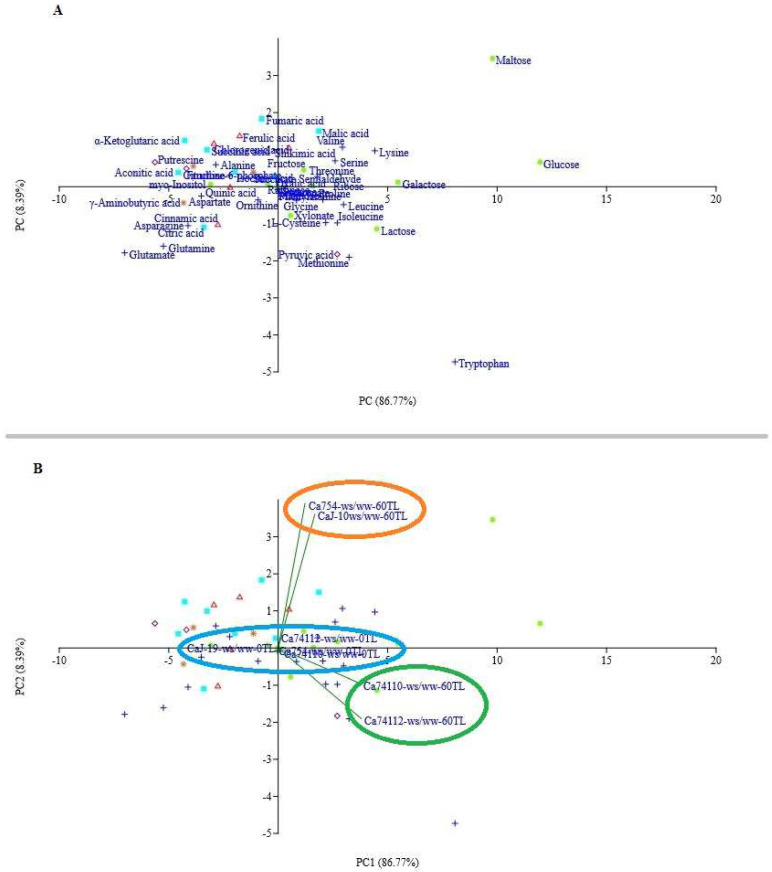
Principle component analysis (PCA) plot (x—first component, y—second component) with (**A**) the row labels of each metabolite and (**B**) the biplot of the *Ca*754, *Ca*J-19, *Ca*74110, and *Ca*74112 coffee genotypes, based on GC/MS analysis, at 0 and 60 days after drought initiation. Amino acids (blue pluses), sugars (orange dots), and glycolysis (pink diamonds), tricarboxylic acid (aqua blue squares), γ-aminobutyric acid (brown stars), and shikimic (red triangles) pathway components.

## Data Availability

The relevant data applicable to this research are provided within the paper and are also available on request from the corresponding author.

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
