# Peer review of "Drought Stress Responses in Arabica Coffee Genotypes: Physiological and Metabolic Insights"

_plants, 2024, doi:10.3390/plants13060828_

Round 1

Reviewer 1 Report

Comments and Suggestions for Authors

Excellent work with low variability.

Fig 1 term adult coffee genotypes , might better be termed seedlings as plants are less than a year old and have not flowered.

Measurements were based upon above ground organs-stem and leaves, at the minimum might not root mass even soil water content at 60 DADB be of interest?

Not evident that you identified an enzyme to modify much less a gene. You have identified metabolites and pathways in seedlings that might permit more rapid screening of progeny of drought tolerant x high yield & quality crosses in the breeding program

Supplemental figures should include photos of the 4 genotypes at O and 60 DADB as ww and ws.

Author Response

Dear Reviewer,

Attached herein, please see the attachment of our review response.

Regards,

Reviewer 2 Report

Comments and Suggestions for Authors

1.     I suggest the results section could be condensed into three parts: phenotype, physiology, and metabolism.

2.     Writing for the results section is too long-winded, such as 3.6, 3.7 et al.

3.     Figure 5 could be replaced by a heatmap.

Author Response

Dear Reviewer,

Attached herein, please see the review response.

Regards,

Reviewer 3 Report

Comments and Suggestions for Authors

This study explores physiological and metabolite changes in different coffee genotypes exhibiting varying degrees of drought tolerance—specifically, the relatively tolerant Ca74110 and Ca74112, as well as the sensitive Ca754 and CaJ-19—under both well-watered conditions and during terminal drought stress periods across two time intervals (0 and 60 days following the initiation of stress). This study provides a theoretical foundation for the development of drought-resistant varieties. Results are interesting and informative for future research. In my opinion, there are still some aspects of this article that need modification. Some suggestions are as below.

1. Some figures lack legends; for instance, in Figure 2, it should be indicated what each different color represents.

2. The article examines physiological indicators of different drought-tolerant plant types after drought stress, and additionally presents the cultivation process of plant materials (Figure S1-S3). However, I suggest incorporating phenotypic images of  Ca754, CaJ-19, Ca74110, and Ca74112, under well water (ww) and drought stress conditions (ws), which would enhance the readability and completeness of the article.

3. The description of the results of the significant difference analysis in the article is quite confusing. For instance, in the results section 3.2, it states, “In well-watered conditions, there were no significant (p<0.05) differences among......at the end of the experiment, the mean relative water content was significantly (p<0.05) lower than......” Does the use of 'p<0.05' indicate both significant and nonsignificant differences throughout the entire text? It is recommended to ensure a consistent and accurate description throughout the entire article.

Comments on the Quality of English Language

Minor editing of English language required.

Author Response

(The authors gave the same response as above.)

Round 2

Reviewer 2 Report

Comments and Suggestions for Authors

I have no further comments.